# Multiple Red Flags of Cardiac Amyloidosis in a Single Patient: Clinical Manifestations of an Underdiagnosed Disease

**DOI:** 10.3390/diagnostics14242812

**Published:** 2024-12-13

**Authors:** Emil Julian Dąbrowski, Wiktoria Urszula Kozłowska, Patrycja Oliwia Lipska, Urszula Matys, Szymon Pogorzelski, Marcin Kożuch, Sławomir Dobrzycki

**Affiliations:** Department of Invasive Cardiology, Medical University of Białystok, 15-089 Białystok, Poland

**Keywords:** cardiomyopathy, amyloidosis, scintigraphy, heart failure

## Abstract

Cardiac transthyretin amyloidosis is an underdiagnosed disorder with significant diagnostic difficulties due to its non-specific clinical manifestations. It is caused by the deposition of protein aggregates with an abnormal tertiary structure in the extracellular matrix. Their accumulation leads to the development of hypertrophic and restrictive cardiomyopathy and, at a later stage, heart failure with preserved ejection fraction syndrome. Depending on the pathogenesis, there are different types of the disease—hereditary and age-related wild-type transthyretin amyloidosis. We present the case of an 85-year-old woman who was referred to the department with a two-month history of exertional dyspnea in New York Heart Association functional class II. After reviewing the initial findings, several red flags for cardiac amyloidosis (CA) were identified. Following the diagnostic algorithm, scintigraphy was performed and showed significant radioisotope accumulation in the myocardium, confirming the suspected disease. In this manuscript, we present the current recommendations and diagnostic pathway, discussing in detail both available and emerging treatment options. As early diagnosis is essential to prevent the development of serious complications, we would like to highlight the pitfalls in diagnosing CA and emphasize the need to be aware of its variable clinical presentation and red flags.

**Figure 1 diagnostics-14-02812-f001:**
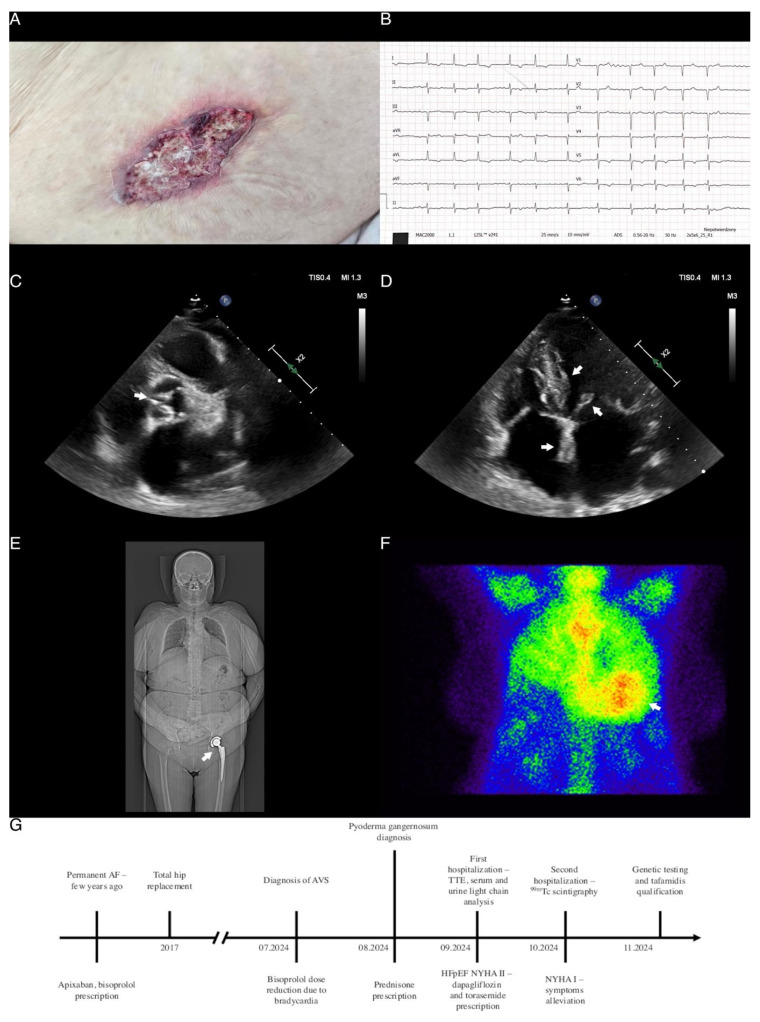
Clinical and radiological red flag findings in patient with transthyretin amyloidosis. (**A**) Pyoderma gangrenosum ulceration of the abdomen. (**B**) Electrocardiogram showing atrial fibrillation with a mean QRS rate of 78 beats per minute, normal axis deviation, and inverted T waves in multiple leads. Note the relatively low voltage as compared to the severely thickened left ventricular walls, as shown in the following panels. (**C**) Transthoracic echocardiography (TTE), parasternal short-axis view, arrow showing moderate aortic stenosis with leaflet thickening and a hemodynamically assessed aortic valve area of 1.4 cm^2^. (**D**) TTE, apical four-chamber view, arrows showing increased right and left ventricular wall thickness with a granular appearance, interatrial septum and mitral leaflet thickening and enlarged atria. (**E**) Low-dose whole-body computed tomography scan, arrow showing left hip acetabular and femoral components after total hip replacement. (**F**) Scintigraphy with ^99m^Tc-hydroxymethylene diphosphonate (HMDP) showing grade 3 myocardial uptake. (**G**) Schematic timeline presenting patient’s signs, performed diagnostic tests and prescribed treatment. An 85-year-old woman with suspected aortic stenosis, permanent atrial fibrillation, and heart failure with preserved ejection fraction (HFpEF) was referred to the Department due to a two-month history of exertional shortness of breath classified as New York Heart Association (NYHA) functional class II. Notably, two weeks prior to hospitalization, the patient had discontinued antihypertensive treatment, which partially alleviated dyspnea. Past medical history was notable only for left hip alloplasty several years ago (Panel E). A physical examination revealed a systolic murmur at the aortic listening post, mild peripheral edema, and pyoderma gangrenosum ulcerations in the right popliteal fossa and abdomen (Panel A). Laboratory tests showed elevated NT-proBNP concentrations (2728.7 pg/mL) and persistently elevated high-sensitivity cardiac troponin levels (32 ng/L). An ECG showed atrial fibrillation with a mean QRS rate of 78 per minute, a normal axis, and negative T waves in leads I, aVL, V5-V6 (Panel B). Echocardiography revealed significant left ventricular hypertrophy (interventricular septum thickness of 16 mm) with a granular appearance, dilatation of both atria, moderate aortic valve stenosis (AVS), low cardiac output with left ventricle hyperkinesis, and an estimated ejection fraction of 80% (Panels C and D).After reviewing the initial findings, multiple red flags for cardiac amyloidosis (CA) were identified, including AVS, HFpEF, echocardiographic and electrocardiographic abnormalities, persistently elevated cardiac troponin levels, and a history of hip alloplasty [1,2,3]. Further testing showed mildly elevated urine free lambda chains and serum IgA concentrations, which, according to a hematology consultant, were related to the chronic pyoderma gangrenosum ulcerations, ruling out light chain CA. In light of clinical features highly suggestive of CA and following the diagnostic algorithm [1], scintigraphy with a 99mTc tracer was scheduled. Due to dyspnea and signs of fluid overload, treatment with dapagliflozin and torasemide was initiated. Three weeks later, during a second hospitalization, scintigraphy was performed, revealing significant (grade 3) radioisotope accumulation in the myocardium, confirming transthyretin CA (Panel F). Genetic testing to differentiate between hereditary and wild-type transthyretin CA was scheduled, and the patient was referred to a CA-specialized center for further evaluation and qualification for tafamidis treatment (Panel G) [2]. CA is widely underdiagnosed, with the prevalence estimated to be as high as 8% and 16% in patients with HFpEF and AVS, respectively [4]. Diagnostic difficulties arise due to the non-specific, multiorgan manifestations caused by the deposition of protein aggregates in the extracellular matrix as a result of abnormal tertiary protein structure (Appendix A) [1,2,3,5]. Cardiac accumulation leads to the simultaneous development of hypertrophic and restrictive cardiomyopathy and, at a later stage, HFpEF [5]. Currently, only tafamidis—a stabilizer of the native transthyretin tetrameric structure [1,2,5]—is available as a first-line treatment for both hereditary and age-related wild-type transthyretin CA. Tafamidis has been shown to reduce cardiovascular hospitalizations, mortality, and improve quality of life and functional capacity [1,2]. The recently published HELIOS-B trial demonstrated a reduction in the risk of death from any cause and cardiovascular events in patients with CA treated with vutrisiran—a drug already approved in the European Union and the United States for the treatment of polyneuropathy caused by hereditary transthyretin amyloidosis [6]. Importantly, the reported effects were consistent across subgroups, regardless of concurrent tafamidis treatment. This case highlights the pitfalls in diagnosing CA, emphasizing the need for greater awareness of its varied clinical presentations and red flags. As novel therapies continue to emerge, early diagnosis is essential for the initiation of targeted treatment, preventing the development of severe complications.

## Data Availability

No new data were created or analyzed in this study.

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
