# Peer review of "Multiple Red Flags of Cardiac Amyloidosis in a Single Patient: Clinical Manifestations of an Underdiagnosed Disease"

_diagnostics, 2024, doi:10.3390/diagnostics14242812_

Round 1
Reviewer 1 Report
Comments and Suggestions for Authors
The manuscript presents a case from real clinical practice that may be of interest to a broad audience. However, additional information is needed for clarity and completeness.
1. Please clarify why the history of hip alloplasty is categorized as a red flag for amyloidosis.
2. Including a timeline of the patient's signs and treatments would enhance the manuscript and improve its educational value.
3. Kindly add details of the initial and follow-up treatment adjustments.
4. Please provide more information on the reason for the second hospitalization that occurred within three weeks.
5. Include the results of genetic testing, and specify the final form of ATTR.
6. Regarding the reported 8% prevalence in HFpEF, please provide more details about the cohort of HFpEF patients.
7. I recommend listing the other common signs of ATTR that were either present or absent in this case to further enhance the educational aspect of the manuscript.
Comments on the Quality of English Language
English must be improved significantly
Author Response
The manuscript presents a case from real clinical practice that may be of interest to a broad audience. However, additional information is needed for clarity and completeness.
- Please clarify why the history of hip alloplasty is categorized as a red flag for amyloidosis.
Reply 1: Thank you for this comment. We are pleased to provide the following clarification. Hip or knee arthroplasty is regarded as a potential indicator of amyloidosis due to the propensity of transthyretin protein aggregates to deposit in ligaments, tendons, and cartilage, which can result in joint destruction and the necessity for joint replacement. Although non-specific, such orthopaedic manifestations are more prevalent in patients with ATTR-CA than in the general population [1]. We are grateful for the opportunity to elaborate on this point, as it highlights the multidisciplinary nature of amyloidosis diagnosis and the necessity for heightened clinical suspicion.
Changes 1: None
[1] Rubin J, Alvarez J, Teruya S, Castano A, Lehman RA, Weidenbaum M, Geller JA, Helmke S, Maurer MS. Hip and knee arthroplasty are common among patients with transthyretin cardiac amyloidosis, occurring years before cardiac amyloid diagnosis: can we identify affected patients earlier? Amyloid. 2017 Dec;24(4):226-230. doi: 10.1080/13506129.2017.1375908. Epub 2017 Sep 14. PMID: 28906148.
- Including a timeline of the patient's signs and treatments would enhance the manuscript and improve its educational value.
Reply 2: Thank you very much for this suggestion. Now we have prepared a timeline with approximate dates of events, prescribed treatment and symptoms.
Changes 2: Please see timeline now included as Figure 1 G.
- Kindly add details of the initial and follow-up treatment adjustments.
Reply 3: As the reviewer has suggested, now we have added information regarding treatment adjustments in the main text and in updated Figure 1.
Changes 3: Please see main manuscript updated and Figure 1 modified.
- Please provide more information on the reason for the second hospitalization that occurred within three weeks.
Reply 4: Thank you for this comment. We are glad to clarify. In our institution, scintigraphy with the 99mTc-HMDP tracer is not routinely performed. Therefore, after consulting the Department of Nuclear Medicine, the second hospitalization was scheduled at the earliest possible date.
Changes 4: None.
- Include the results of genetic testing, and specify the final form of ATTR.
Reply 5: Thank you for this comment. Unfortunately, the patient was recently visiting the outpatient clinic at our institution and is still awaiting the genetic testing results, which are expected in early 2025.
Changes 5: None.
- Regarding the reported 8% prevalence in HFpEF, please provide more details about the cohort of HFpEF patients.
Reply 6: Thank you for raising this very important topic. Cardiac deposition of protein aggregates in the extracellular matrix as a result of abnormal tertiary protein structure leads to the simultaneous development of hypertrophic and restrictive cardiomyopathy and, at a later stage, HFpEF.
The prevalence of cardiac amyloidosis in heart failure with preserved ejection fraction patients, reported to be 8%, is based on study by Healy et al. Studied cohort was predominantly composed of older individuals and a higher proportion of male patients. The study cohort consisted of 86 individuals aged 60 years or older who had been diagnosed with heart failure with preserved ejection fraction (HFpEF) and exhibited New York Heart Association (NYHA) Class II–IV symptoms. These patients were prospectively screened between July 2019 and July 2022. Participants were excluded if they had a history or current diagnosis of heart failure with reduced ejection fraction (HFrEF, defined as EF ≤50%), unstable arrhythmia, cardiogenic shock, recent acute coronary syndrome (within the last three months), severe valvular heart disease, or a known monoclonal gammopathy such as multiple myeloma, immunoglobulin light chain (AL) amyloidosis, or monoclonal gammopathy of undetermined significance (MGUS). Additionally, individuals with previous evaluation for cardiac amyloidosis were excluded. At the time of their diagnosis, these patients demonstrated significant left ventricular wall thickening and elevated levels of high-sensitivity troponin, indicating myocardial involvement and underscoring the diagnostic challenges in this population [1].
We are grateful for the opportunity to expand on these details, as they highlight the subtle diagnostic considerations that are often overlooked in this under-recognised population.
Changes 6: None.
[1] Healy L, Giblin G, Gray A, Starr N, Murphy L, O'Sullivan D, Kavanagh E, Howley C, Tracey C, Morrin E, McDaid A, Clarke A, O'Neill JO, Joyce E, O'Connell M, Mahon NG. Prevalence of transthyretin cardiac amyloidosis in undifferentiated heart failure with preserved ejection fraction. ESC Heart Fail. 2024 Nov 7. doi: 10.1002/ehf2.15112. Epub ahead of print. PMID: 39508367.
- I recommend listing the other common signs of ATTR that were either present or absent in this case to further enhance the educational aspect of the manuscript.
Reply 7: Thank you for this suggestion. As the Reviewer has suggested, we have now included table summarizing common signs which may be seen in patients with amyloidosis. We bolded information on findings presented in the current paper.
Changes 7: Please see Table 1. now included in the manuscript.
Comments on the Quality of English Language: English must be improved significantly
Reply: Thank you for this comment. As suggested, the entire manuscript has been thoroughly revised language-wise.
Changes: Please see multiple changes throughout the manuscript.
Reviewer 2 Report
Comments and Suggestions for Authors
The authors present a case of cardiac amyloidosis, with the images illustrating some of the red flags seen in this disease.
I think the paper would be much improved if the authors added a table with all the common red flags of cardiac amyloidosis.
Also, in my opinion, scintigraphy is not a red flag, it is a diagnostic criterion.
Author Response
The authors present a case of cardiac amyloidosis, with the images illustrating some of the red flags seen in this disease.
I think the paper would be much improved if the authors added a table with all the common red flags of cardiac amyloidosis.
Reply: Thank you for this great suggestion. We have added Table 1. summarizing key red flags which may occur in patients with amyloidosis.
Changes: Please see Table 1. now included in the manuscript.
Also, in my opinion, scintigraphy is not a red flag, it is a diagnostic criterion.
Reply: We indeed agree with the Reviewer – scintigraphy is a key method used during diagnostic algorithm in patients with suspected amyloidosis. As the manuscript presents single patient with multiple red flags, we intended to describe the diagnostic pathway, which was applied to set the diagnosis. As some of the phrases in manuscript might have been ambiguous, we have now rephrased them to make a distinction between red flags and diagnostic work-up.
Changes: Please see modified paragraph regarding scintigraphy findings.